# Study of High-Silicon Steel as Interior Rotor for High-Speed Motor Considering the Influence of Multi-Physical Field Coupling and Slotting Process

**DOI:** 10.3390/ma15238502

**Published:** 2022-11-29

**Authors:** Deji Ma, Baozhi Tian, Xuejie Zheng, Yulin Li, Shibo Xu, Ruilin Pei

**Affiliations:** 1Department of Electric Engineering, Shenyang University of Technology, Shenyang 110870, China; 2Suzhou Inn-Mag New Energy Ltd., Suzhou 215000, China

**Keywords:** motor, interior rotor, high-silicon steel, multi-physical field, slotting

## Abstract

Currently, high-speed motors usually adopt rotor structures with surface-mounted permanent magnets, but their sheaths will deteriorate performance significantly. The motor with interior rotor structure has the advantages of high power density and efficiency. At the same time, high silicon steel has low loss and high mechanical strength, which is extremely suitable for high-speed motor rotor core material. Therefore, in this paper, the feasibility of using high silicon steel as the material of an interior rotor high-speed motor is investigated. Firstly, the magnetic properties of high silicon steel under multi-physical fields were tested and analyzed in comparison with conventional silicon steel. Meanwhile, an interior rotor structure of high-speed motor using high silicon steel as the rotor core is proposed, and its electromagnetic, mechanical, and thermal properties are simulated and evaluated. Then, the experimental comparative analysis was carried out in terms of the slotting process of the core, and the machining of the high silicon steel rotor core was successfully completed. Finally, the feasibility of the research idea was verified by the above theoretical analysis and experimental characterization.

## 1. Introduction

In recent years, the need for more efficient electrical applications is increasing due to the growing environmental concerns and the consequent gradual transition to a decarbonized society [1]. High-speed permanent magnet motors are used in a variety of applications due to their high efficiency and power density, such as aircraft generators, flywheel energy storage systems, high-speed spindles, turbomolecular pumps, air compressors, blowers, turbochargers, and microturbines. However, a large centrifugal force is generated when the rotor runs at high speed, which can cause irreversible mechanical damage to the motor rotor and thus affect the reliable operation of the motor [2]. In order to avoid the influence of centrifugal forces, a sleeve structure is usually applied on the outside of the motor rotor, currently the mainstream sleeve materials are metal and carbon fiber. However, with metal sleeves, eddy currents in the sleeves produce large eddy current losses, which can have a deteriorating effect on the efficiency of the motor. High eddy currents may also lead to overheating of the rotor to the point of irreversible demagnetization of the permanent magnets. If carbon fiber winding is used to protect the rotor from centrifugal forces, it results in a relatively large motor air gap length which affects the performance of the motor. Meanwhile, the heat dissipation capacity of the rotor is reduced, again leading to irreversible demagnetization of the permanent magnets [3,4,5,6]. These problems can be avoided by the rotor’s interior permanent magnet design with no sleeve structure [7].

During the operation of the motor, the centrifugal force affects the rotor structure, which requires the material of the interior rotor to have good mechanical strength while considering the magnetic properties [8]. Kasai analyzed the mechanical strength and magnetic properties of a high-silicon steel (10JNEX900) and found that both properties of this type of steel are superior to those of conventional steels, which makes it compatible with the requirements imposed by interior permanent magnet motor rotors [9]. This study used high silicon steel as the rotor material of a high-speed motor and demonstrated the feasibility of its application [10]. Figure 1 shows the yield strength and iron loss of several materials. Although 1K101 amorphous material has high yield strength and low loss properties, its performance is severely affected by processing and cannot be mass-produced [7]. Materials, such as Vacoflux48 and Vacoflux17, have low yield strengths and are not suitable for rotor core materials of high-speed motors. Compared to high strength steel 20SW1200H and high silicon steel two materials in mechanical properties and loss performance, traditional silicon steel also does not have any advantage. Although the high-strength steel 20SW1200H and high silicon steel 10JNEX900 performance are similar, their mechanical properties and loss performance still have gaps.

Guangwei Liu’s team studied that during motor operation, the silicon steel is subjected to centrifugal force and a certain temperature rise, which causes a certain change in the magnetic properties of the rotor core [11]. Andreas Krings and Oskar Wallmark et al. studied the loss of iron cores as affected by temperature [12] and Junquan Chen et al. developed a model for silicon steel loss considering the effect of temperature [13]. Later, a large number of experiments were conducted, and the temperature coefficients of the loss model were corrected according to the results to verify the correctness of the model [14,15]. Above studies show that the core of the motor is under the operating conditions of coupled multi-physical fields, such as temperature, stress, and electromagnetism, but there is no literature focusing on the magnetic performance of high silicon steel under multi-physical coupled fields.

In addition to the above, the machining process of the motor core also has a non-negligible impact on the performance of the motor [16]. Meanwhile, the literature [17] also points out that the selection of the motor rotor material should determine not only the most suitable material to withstand the stress, but also the best processing technique for the rotor material, as well as the main factors affecting the fatigue resistance of the selected material. Yingzhen Liu’s study showed that the core processing process can have some deteriorating effects on the performance of conventional silicon steel cores [18]. Design with multiple slots inside the rotor should consider the impact of the cutting process on the core material more than the surface-mounted design with a sleeve. The stamping process has the advantages of low cost and high efficiency and is the primary choice for mass production of electric motors. Laser cutting is mainly used for the process of some special purpose motors [16]. The internal microstructure of the cutting part was observed by Aroba Saleem et al. [19] and its effect was studied in [20,21]. In addition, wire electrical discharge machining is also suitable for the processing of high silicon steel cores due to its precise machining accuracy and low deterioration effect on the core. Studying the effect of core processing on high silicon steel cores to find out the suitable slotting for interior rotors will also affect the performance of motors to some extent.

In Section 2 of this paper, the mechanical properties of high silicon steel (10JNEX900) and high strength steel (20SW1200H) of the same thickness are tested. A set of “electric-magnetic-thermal-stress” multi-physical field coupling magnetic property testing devices was built to investigate the different advantages of two materials, as well as to investigate and analyze their magnetic property change law under the multi-physical coupling field. In Section 3, a 90-kW high-speed interior permanent magnet motor is designed, the effect of the two materials on the motor performance under the rotor core of the high-speed motor is calculated by simulation, and the performance differences of electromagnetic, stress, and temperature are compared and analyzed. Section 4 compares the performance of the cores processed by the two different cutting processes and explores the extent of their influence on the core performance. Moreover, we analyze the mechanism related to this influence, then conclude the most suitable cutting process for high silicon steel. Finally, the feasibility of high-silicon steel for high-speed motor rotor is verified by analyzing high-silicon steel rotor in three different dimensions.

## 2. Material Electromagnetic Performance Evaluation

High-Si steels are Fe-Si alloys with a Si mass fraction of 6.5%, where the Si atoms are uniformly diffused inside the material by chemical vapor deposition. The resistivity of the silicon steel material is greatly enhanced by the addition of more Si elements. High-silicon steel sheet has the advantages of low iron consumption, low hysteresis, and high magnetic permeability and is suitable for use as an iron core in electric motors.

High-silicon steel is prepared by chemical vapor deposition (CVD) method, and the process is shown in Figure 2. The SiCl_4_ gas is used to react with the silicon steel strip at high temperature to form a layer of Fe_3_Si, and then the furnace temperature is increased to diffuse Si into the interior of the strip, and the CVD deposition is followed by high temperature diffusion annealing under reducing atmosphere protection to achieve a composition of 6.5% Si [22].

### 2.1. Experiment Method

In this experiment, the magnetic properties of two kinds of silicon steel sheets, 6.5% Si and 20SW1200H, were tested under the conditions of temperature-tension stress-electromagnetic field coupling, using a three-part coupling of silicon steel sheet electromagnetic characteristics test module, mechanical property test module and test environment adjustment module. The principle of the multi-physics field coupled test system is shown in Figure 3.

The experimental sample in this study is a non-standard circular sample, the length of the magnetic circuit of the sample is set to 625 mm, the primary winding and secondary winding are both 200 turns, and the cross-sectional area of the test sample is calculated based on the theoretical thickness and the lamination factor. The test specimen is fixed to the instrument by the auxiliary tooling. The top and bottom of the sample are laminated with oriented silicon steel which has high flux density and low iron loss, and the loss effect on the two wide sides of the sample is thus negligible. The physical diagram of test specimen and device is shown in Figure 4.

### 2.2. Effect of Temperature on Magnetic Properties

Figure 5 demonstrates the effect of temperature on the magnetic properties of the two silicon steel materials under no stress. The saturation flux density of 10JNEX900 at 50 Hz shows a clear tendency to decrease with increasing temperature. The magnetic permeability of the material is maximum at 0 °C. Its permeability also decreases with increasing temperature. The magnetic flux density of 10JNEX900 at 50 Hz and 1000 A/m decreased by 5.07% when the temperature increased from 0 °C to 100 °C. On the contrary, the saturation flux density of 20SW1200H is less affected by temperature, and its permeability also changes slightly with the increase in temperature. When the magnetic field strength is larger than 250 A/m, the saturation flux density shows a decreasing trend with the increase in temperature. The flux density of 20SW1200H at 50 Hz and 1000 A/m decreases by 1.8% when the temperature increases from 0 °C to 100 °C. The reason for this phenomenon is mainly due to the continuous increase in the speed of molecular motion as the temperature increases. This phenomenon prevents the movement of magnetic domains and domain walls during the magnetization process, leading to a decrease in the magnetic permeability of the material, which in turn reduces the saturation magnetic flux density of the material [23].

In terms of iron loss, the loss of 10JNEX900 at 400 Hz shows a significant increasing trend with the increase of temperature, while the loss of 20SW1200H shows a slight decreasing trend with the increase of temperature. When the temperature increases from 0 °C to 100 °C, the loss of 10JNEX900 at 400 Hz and 1 T increases by 42.7%, and the loss of 20SW1200H at 400 Hz and 1 T decreases by 2.01%. The loss of 20SW1200H decreases with temperature mainly because the resistivity of the material increases with the temperature, so the eddy current loss decreases too. The opposite result of 10JNEX900 is due to its special preparation process (CVD). When the temperature increases, the internal stress inside the material will lead to an increase in hysteresis loss, and the increase in hysteresis loss is greater than the decrease in eddy current loss, so the total loss of the material increases [22].

### 2.3. Effect of Stress on Magnetic Properties

Figure 6 demonstrates the effect of tensile stress on the magnetic properties of the two silicon steel materials at 25 °C. The saturation flux density of 10JNEX900 at 50 Hz shows a decreasing trend with the increase of tensile stress, but the curves at 1000 A/m attachment 25 Mpa and 50 Mpa are crossed. The permeability of 10JNEX900 is almost unaffected by the tensile stress, while the saturation flux density and permeability of 20SW1200H are both greatly affected by increasing tensile stress, and the saturation flux density decreases with higher tensile stress. The magnetic flux density of 10JNEX900 at 50 Hz and 1000 A/m decreased by 3.6% when the stress was increased from 0 Mpa to 75 Mpa.

The loss of 10JNEX900 at 400 Hz shows a significant increasing trend with the increase of tensile stress, while the loss of 20SW1200H shows a significant decreasing trend with the increase of temperature. When the stress increases from 0 Mpa to 75 Mpa, the loss of 10JNEX900 at 400 Hz and 1 T increases by 17.58%, and the loss of 20SW1200H at 400 Hz and 1 T decreases by 16.52%. The main reason for this phenomenon in 10JNEX900 is the presence of stresses that hinder the movement of the magnetic domains and the walls of the domains increasing the energy loss and reducing the saturation flux density. In contrast, the high yield strength capability of high-strength silicon steels is obtained partly by the thermal expansion and infiltration process with lower dislocations, which induce stronger residual internal stresses and thus prevent the formation of 180° magnetic domains under tensile stress [8]. Therefore, the loss of high-strength silicon steel 20SW1200H is reduced.

### 2.4. Effect of Temperature-Stress Coupling on Magnetic Properties

Figure 7 shows the variation of saturation flux density and loss of 10JNEX900 under the condition of multi-physics field coupling. The magnetic properties of 10JNEX900 vary with temperature and tensile stress as analyzed above. The lowest point of its magnetic flux density occurs at 100 °C and 75 Mpa, and the lowest point of loss occurs at 0 °C and 0 Mpa. The flux density and loss of 10JNEX900 increased by 8.42% and 98.21% from the lowest point to the highest point, respectively. Therefore, it can be concluded that the deterioration of material loss is more obvious in the case of coupled temperature and tensile stress.

## 3. Motor Finite Element Analysis

In order to evaluate the performance of high-silicon steel 10JNEX900 as an interior rotor for high-speed motors, this section analyzes the performance of rotors made from 10JNEX900 and 20SW1200H in terms of electromagnetic, mechanical, and thermal properties utilizing simulation calculations.

### 3.1. Main Structural Parameters of Motor

Figure 8 illustrates the structure of the permanent magnet synchronous motor proposed in this paper. The motor’s pole-slot ratio is 8 poles and 48 slots, and the rotor’s permanent magnet structure is designed as an interior U-shape, with a maximum motor speed of 19,000 rpm. The main performance parameters of the motor are shown in Table 1.

### 3.2. Motor Electromagnetic Performance Analysis

Figure 9 shows the flux density distribution for the two material motors. In this section, the electromagnetic performance of two motors is analyzed using the finite element analysis software Maxwell. The results show that the flux density at the rotor spacer bridge of the 10JNEX900 motor is slightly lower than that of the 20SW1200H motor. Figure 10 shows the loss distribution for the two material motors under loaded condition. Although the rotor loss of the PM synchronous motor is lower, it can still be seen that the loss at the outer edge of the rotor of the 10JNEX900 motor is slightly lower than that of the 20SW1200H motor. All of the above reasons are because the 10JNEX900 has a larger saturation flux density and permeability and has lower losses compared to the 20SW1200H, but with less impact. This will give the motor a higher operating efficiency.

Figure 11 shows the back EMF waveforms of the two motors during no-load operation. The difference between the two no-load counter potential cases is very small. This indicates that for the same motor, the change in rotor material has less effect on the motor’s no-load back EMF.

Figure 12 shows the load air-gap flux density waveforms for both motors. The overall difference between the two is small, but the air-gap flux density of the 10JNEX900 motor is greater than that of the 20SW1200H motor when the electrical angle is between 100° and 250°. This is mainly due to the above-mentioned reasons. The 10JNEX900 has better magnetic properties compared to 20SW1200H. As a result, the 10JNEX900 motor also has better overload performance.

The MAP comparison of motor efficiency for the two rotor core materials is shown in Figure 13. There is a difference in the maximum operating efficiency and the percentage of high efficiency area between the two motors. However, when the rotor core is 10JNEX900, the maximum operating efficiency of the motor operation reaches 98.1%, which is 0.1% higher compared to 98.06% of 20SW1200H. The area of high-efficiency zones with efficiency >90% also increased from 92.1% to 92.43%. Therefore, compared to the conventional high-strength silicon steel 20SW1200H, 10JNEX900 has a greater advantage in terms of motor operating efficiency in the case of finite element analysis.

### 3.3. Rotor Mechanical Stress Analysis

In order to verify whether the rotor can safely withstand the centrifugal force caused by the maximum speed of the motor, the centrifugal force of the motor with two different rotor materials at maximum speed was analyzed in this paper using ANSYS software, as shown in Figure 14. It is obvious that the maximum stress points on the rotor are at the magnetic isolation bridge. The maximum stress value of 10JNEX900 motor is 451.43 Mpa, while the yield strength of 10JNEX900 is 570 Mpa. With a safety margin of 1.2 times considered, the 10JNEX900 fully meets the requirements in terms of mechanical performance. The maximum stress value of 20SW1200H motor is 451.75 Mpa, while the yield strength of the material is 480 Mpa. Therefore, for the same structure of the rotor, 20SW1200H can meet the requirements of the motor in terms of mechanical performance, but there is no outflow of sufficient safety margin. Since the mass density of the two materials is different, the distribution of stresses in the 10JNEX900 rotor is lower for the same rotor structure.

### 3.4. Rotor Temperature Rise Performance Analysis

Because this motor runs at high power and speed condition, the temperature rise of the motor rotor will also be higher. In the above loss and efficiency analysis, the 10JNEX900 motor shows greater advantages. The cooling method of the motor is water-cooled. In order to analyze the temperature rise of these two motors at the maximum speed, this paper uses ANSYS software to perform accurate simulations on the 3D model of the motor.

Regarding the details of the simulation analysis, the ambient temperature and the inlet water temperature were set to 30 °C, and the inlet water velocity at the inlet end of the motor housing was 1 m/s. Figure 15 shows the rotor temperature distribution of the motor for both materials at the peak motor speed. It can be clearly seen that the overall temperature distribution of the 20SW1200H motor rotor is higher than that of the 10JNEX900, and the highest temperature point is 3.33 °C higher. This is due to the fact that the 10JNEX900 core produces lower iron consumption when the motor is running at peak speed, so the rotor temperature will be lower. The motor rotor has the highest temperature at the outer diameter edge and the lowest temperature at the inner diameter edge. This is due to the fact that, in addition to the heat generated by the motor rotor itself, the heat generated by the windings is transferred through the air gap to the outer surface of the rotor and further to the inside of the motor rotor. The inner diameter of the motor is at the direct connection of the rotor, and the heat of the rotor will be transferred through the rotor.

## 4. Research of Slotting Method

The magnetization properties of its teeth and yoke are particularly sensitive to the processing stresses generated during the machining process, so there are significant differences between the actual motor performance that is finally exhibited and the calculated values after simulation through the original material data.

Since high silicon steel cores are usually used in stators to reduce core losses, the effect of processing on the mechanical properties of the material is not yet clear. The commonly used core cutting methods are punching, laser cutting, wire electrical discharge machining (WEDM), and water jet cutting [24,25,26]. The deterioration of the magnetic properties by punching is more severe and due to the brittle material of high silicon steel the formability is poor. The process of water jet cutting for a single piece of silicon steel is complicated and its cutting accuracy for the whole core is low [27]. Due to the thinness and brittleness of high silicon steel sheets, they are prone to cracking. Moreover, combined with the complex structure of the motor rotor and its high-speed characteristics, a small defect in the core can lead to rotor cracking at high speeds. To ensure the success of the prototype, experimental studies on the cutting quality of WEDM and laser cutting methods and their effects on magnetic properties were conducted in this paper, and the WEDM method was finally chosen to manufacture the prototype.

### 4.1. Wire Electrical Discharge Machining

The WEDM equipment used in this study is the DK7732 from SUZHOU BAOMA, China. WEDM is the most commonly used core cutting process for prototyping. It has good universality and good cutting quality and is suitable for cutting cores of various new materials and complex shapes. However, the cutting efficiency of WEDM is low, and if the cutting area is large and the structure is complicated, it will lead to longer cutting time. Figure 16 shows a sample of the experimental ring cut by WEDM. It can be seen that the sample surface quality material is smooth, the burr is small, and there is no shattering. The grain between the slices is still clearly visible, and there is no conduction between the stacks. The maximum error of dimensional measurement after CMM is controlled at ±0.02 mm with high accuracy. Because this paper only considers the production of the principle prototype, after the above comprehensive analysis, WEDM is suitable for the production of the prototype in this paper.

### 4.2. Laser Cutting

For ordinary silicon steel sheets, laser cutting is a better choice of grooving method. The GF6025Plus laser cutting machine from HGTECH, China, was used for this study. Laser cutting has high cutting efficiency, its cutting speed can reach 12 m/min, and the cutting accuracy can also be controlled within ±0.03 mm. However, this method is only suitable for cutting single silicon steel sheet. If the stacked cores are cut by laser, it will cause the adhesive and coating between the stacked sheets to be melted at high temperature, resulting in a short circuit on the cutting surface and increasing the iron loss of the sample. Three problems arise when slotting high-silicon steel using this method: (1) Due to the high silicon steel sheet its brittleness is high, single piece cutting process if the operation is not standardized will easily appear single piece shattering situation. (2) Because the high silicon steel sheet is very thin, using laser cutting will cause the whole cutting and stacking process to become complicated. (3) The edge part of high silicon steel sheet processed by laser cutting will produce large burrs, which will lead to the reduction of the stacking coefficient of the iron core and eventually affect the torque performance of the motor. Figure 17 shows a test specimen of the laser cutting process molding. Therefore, it can be concluded from the analysis that high silicon steel is not suitable for grooving by this method.

### 4.3. The Effect of Slotting on the Magnetic Properties of the Material

Figure 18 shows the comparison of the magnetic performance test data of 10JNEX900 at 50 Hz and 400 Hz under two different cutting methods. As seen in Figure 18a, the saturation flux density and permeability of the laser cutting sample are significantly lower than those of the WEDM sample during the whole magnetization process at a frequency of 50 Hz. The magnetic flux density of the WEDM sample was 3.74% higher than that of the laser cutting sample when the magnetic field strength was 6000 A/m. From Figure 18b, it can be seen that the iron consumption of the laser cutting sample is significantly higher than that of the WEDM sample at 400 Hz. When the magnetic flux density was 1 T, the iron consumption of the laser cutting sample was 24.2% higher than that of the WEDM sample.

Figure 19 illustrates the organization of the specimen edge cross-section after two different cutting methods. As seen in Figure 19b, the edge of the specimen after WEDM also has no obvious plastic deformation and is in the form of an arc-shaped oblique strip, but the edge area has been melted by high temperature and a small amount of burr has appeared. Due to the rapid high temperature and cooling process that the sample undergoes during cutting, a small thermal stress is generated inside it. As seen in Figure 19b, there is an obvious burning and melting phenomenon on the edge of the specimen after laser cutting, which is due to the thermal stress generated at the laser cutting temperature of more than 1000 degrees. After cooling, a large area of internal cooling stress is generated in its edge part. This internal stress is detrimental to the magnetic properties of the material and results in laser cutting samples with lower saturation flux density and higher iron losses. Therefore, from the point of view of the effect on the magnetic properties of the material, the magnetic properties of the material after WEDM show a better performance.

After the above analysis, the stator-rotor core of the principle prototype was machined by WEDM, as shown in Figure 20. It can be seen that the laminated core shows a good appearance after cutting, which verifies the thesis that WEDM is more suitable for high silicon steel cutting in the paper.

## 5. Conclusions

This paper presents an interior rotor structure of a high-speed permanent magnet synchronous motor with high silicon steel (10JNEX900) as the rotor material. To further determine the idea that high silicon steel is suitable for high-speed motor rotors, the magnetic properties of 10JNEX900 were measured in this paper. The results show that 10JNEX900 has better magnetic properties than the conventional high-strength steel 20SW1200H in both the normal state and the multi-physical field coupled state.

Meanwhile, considering the actual operating conditions of the high-speed motor rotor, the proposed rotor is analyzed in terms of electromagnetic performance, mechanical performance, and thermal performance, and the analysis results of the 10JNEX900 rotor and the 20SW1200H rotor are compared and analyzed in this paper. The results show that motors with high-silicon steel rotors have higher efficiency, better performance and lower temperature rise than rotors with conventional high-strength steel.

In order to solve the problem of difficult grooving due to the high brittleness of high silicon steel, this paper compares the two most commonly used methods for cutting high silicon steel cores. The results show that WEDM has better cut quality than laser cutting and its effect on the magnetic properties of the material is relatively small. The WEDM method is more suitable for slotting high-silicon steel cores. After the above comprehensive analysis, high silicon steel can be considered as a rotor core material for high-speed motors, which provides a solution to the problem of centrifugal force and loss in the rotor that exists during high speed operation of the motor.

In the future, we will continue the fabrication of the principle prototype and complete the performance test analysis of the whole machine. Further verification of the suitability of high-silicon steel as a rotor for high-speed motors will be performed, and problems arising from the prototype production will be fed back into the design phase for a more in-depth study of this direction.

## Figures and Tables

**Figure 1 materials-15-08502-f001:**
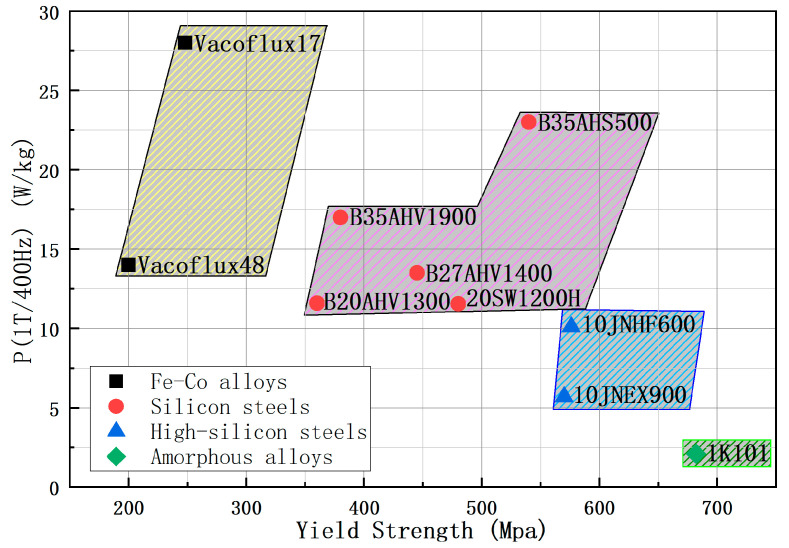
Comparison of yield strength and core loss for different silicon steels.

**Figure 2 materials-15-08502-f002:**
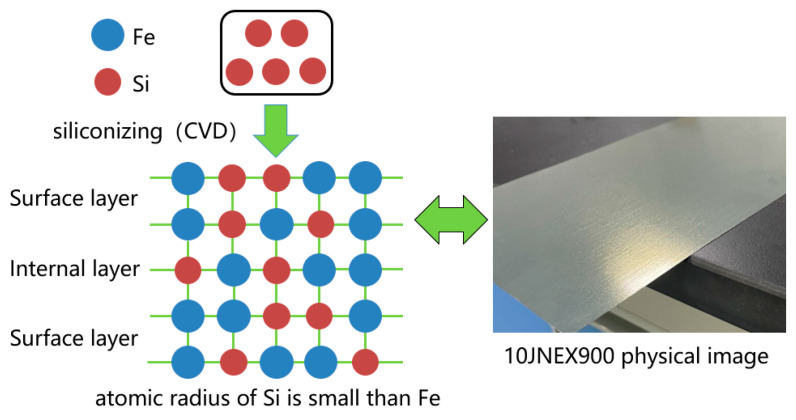
10JNEX900 CVD process and physical schematic diagram.

**Figure 3 materials-15-08502-f003:**
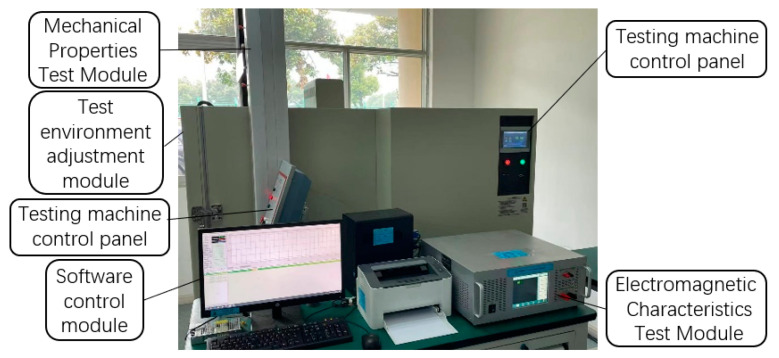
Multi-physics field coupled magnetic energy measurement system.

**Figure 4 materials-15-08502-f004:**
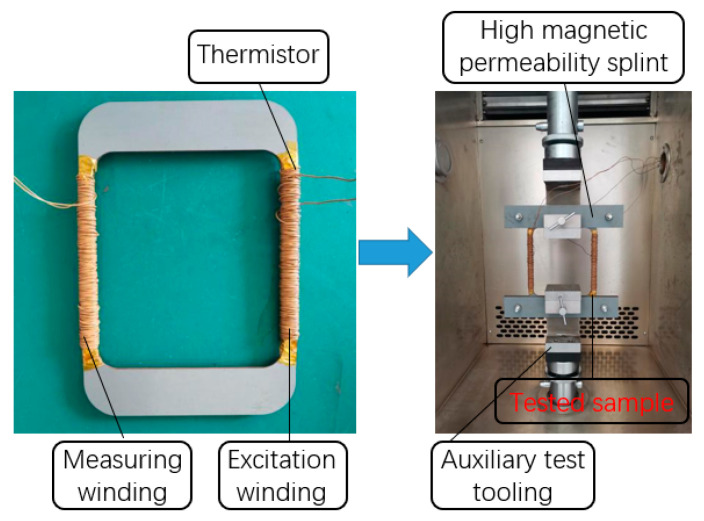
Physical diagram of test specimen and process.

**Figure 5 materials-15-08502-f005:**
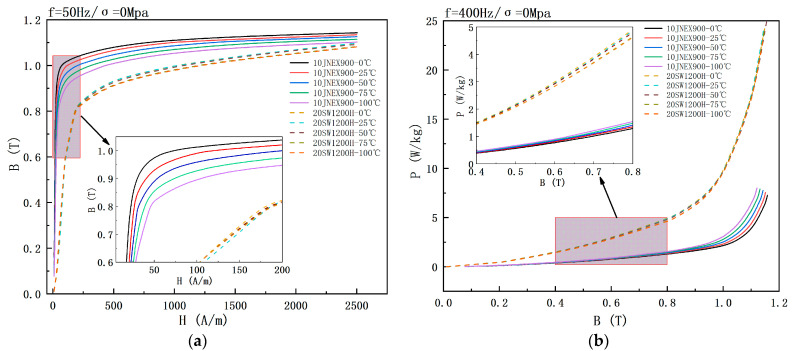
Effect of temperature on magnetic properties of two silicon steels at 0 Mpa. (**a**) B-H curve. (**b**) B-P curve.

**Figure 6 materials-15-08502-f006:**
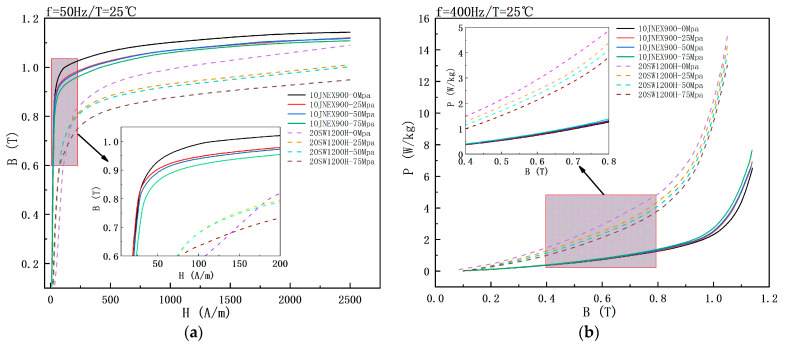
Effect of tensile stress on magnetic properties of two silicon steels at 25 °C. (**a**) B-H curve. (**b**) B-P curve.

**Figure 7 materials-15-08502-f007:**
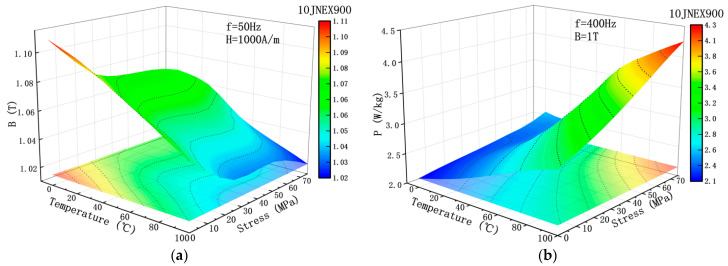
Effect of multi-physical field coupling on the magnetic properties of 10JNEX900. (**a**) Trend of B. (**b**) Trend of P.

**Figure 8 materials-15-08502-f008:**
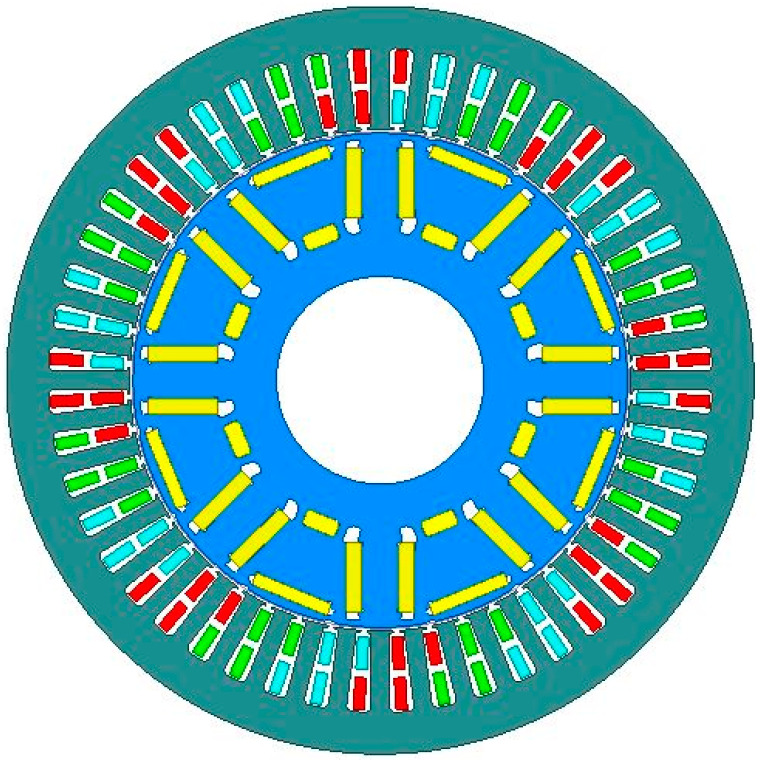
Model diagram of permanent magnet motor.

**Figure 9 materials-15-08502-f009:**
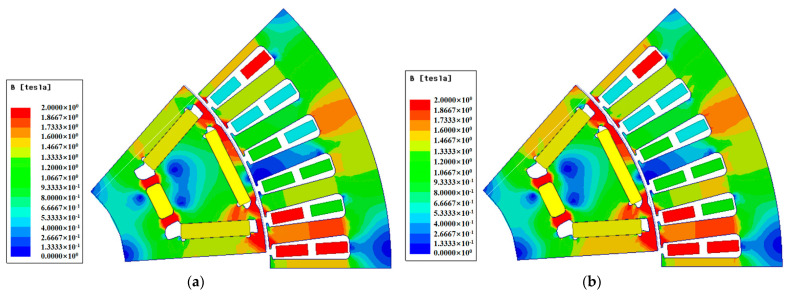
Flux density distribution of two materials motor under loaded condition. (**a**) 10JNEX900. (**b**) 20SW1200H.

**Figure 10 materials-15-08502-f010:**
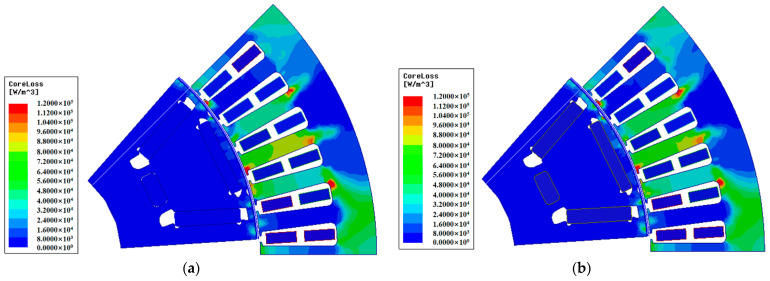
Iron loss distribution of two material motor under loaded condition. (**a**) 10JNEX900. (**b**) 20SW1200H.

**Figure 11 materials-15-08502-f011:**
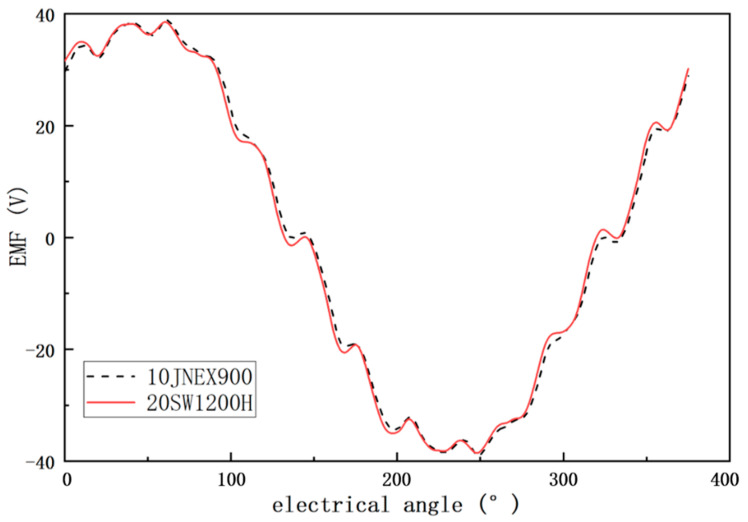
Comparison of no-load back EMF of 10JNEX900 and 20SW1200H motors.

**Figure 12 materials-15-08502-f012:**
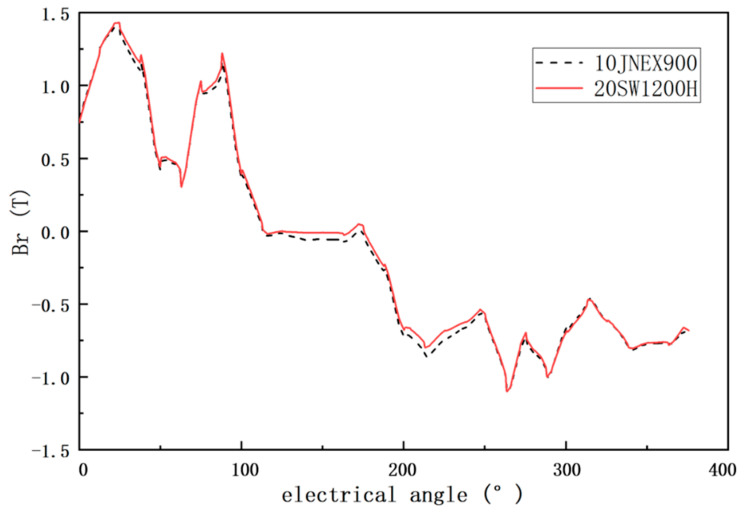
10JNEX900 and 20SW1200H motor load air-gap flux density comparison.

**Figure 13 materials-15-08502-f013:**
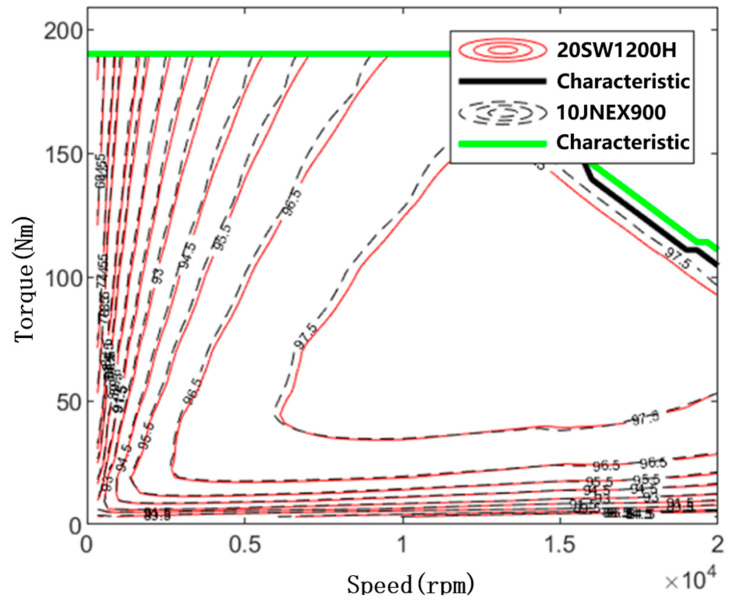
10JNEX900 and 20SW1200H motor efficiency MAP comparison.

**Figure 14 materials-15-08502-f014:**
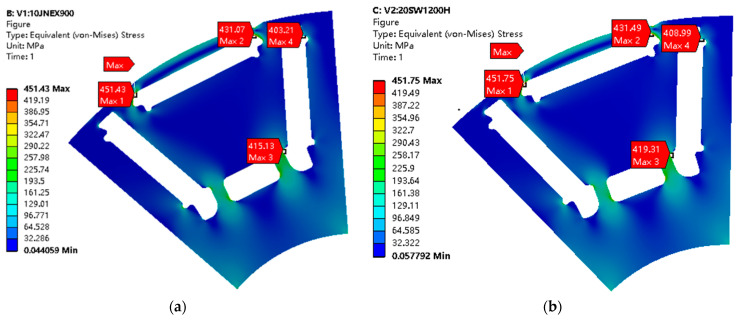
Mechanical stress distribution of rotor cores of two materials at 19,000 r/min. (**a**) 10JNEX900. (**b**) 20SW1200H.

**Figure 15 materials-15-08502-f015:**
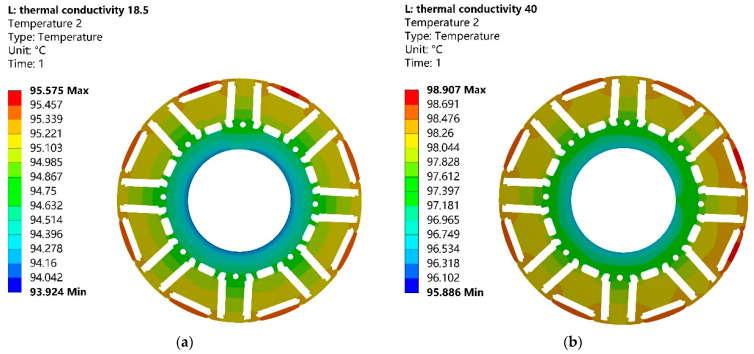
Temperature distribution of motor rotor cores of two materials at 19,000 r/min. (**a**) 10JNEX900. (**b**) 20SW1200H.

**Figure 16 materials-15-08502-f016:**
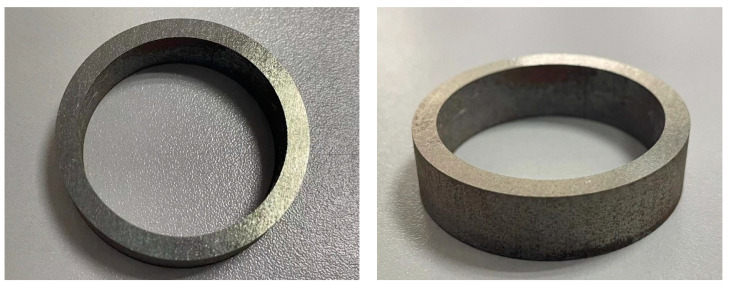
Ring test specimen machined by WEDM.

**Figure 17 materials-15-08502-f017:**
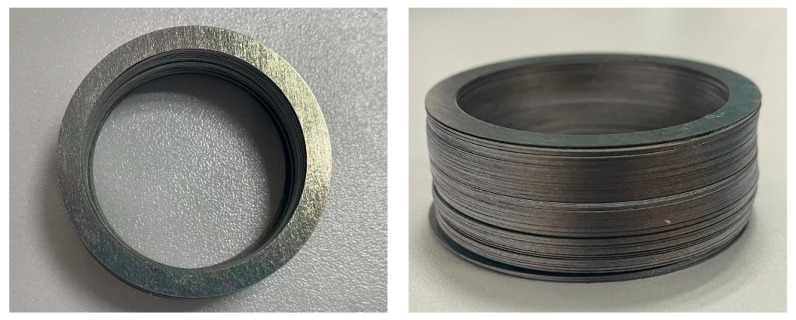
Laser cutting shaped ring test specimen.

**Figure 18 materials-15-08502-f018:**
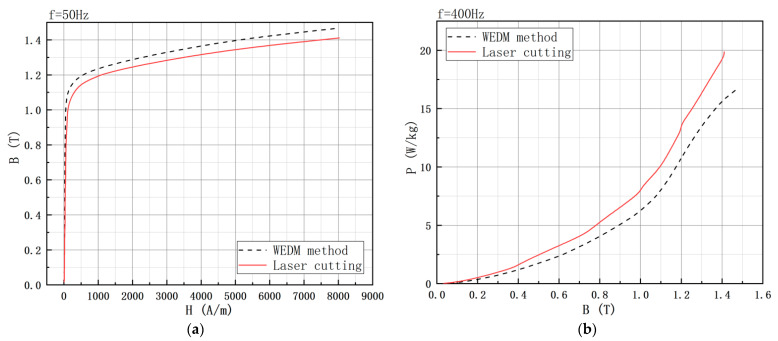
Comparison of magnetic properties of high silicon steel with two different processing methods. (**a**) B-H curve. (**b**) B-P curve.

**Figure 19 materials-15-08502-f019:**
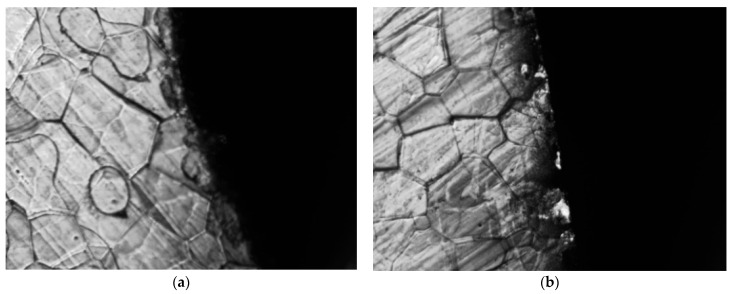
Changes in grain structure at the edges of high silicon steel after grooving in two different ways. (**a**) WEDM. (**b**) Laser cutting.

**Figure 20 materials-15-08502-f020:**
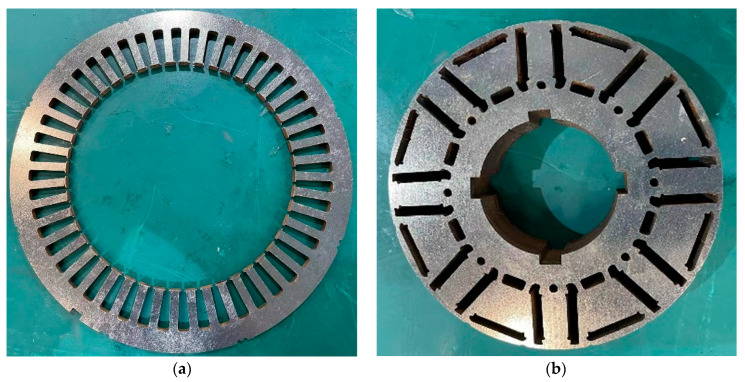
WEDM core of a motor. (**a**) Stator. (**b**) Rotor.

**Table 1 materials-15-08502-t001:** Some performance parameters of permanent magnet motor.

Parameter	Value	Parameter	Value
Rated power/kW	90	Peak power/kW	210
Rated speed/rpm	9549	Peak speed/rpm	19,000
DC bus voltage/V	540	Peak current/A	500

## Data Availability

Not applicable.

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
