# Peer review of "Study of High-Silicon Steel as Interior Rotor for High-Speed Motor Considering the Influence of Multi-Physical Field Coupling and Slotting Process"

_materials, 2022, doi:10.3390/ma15238502_

Round 1

Reviewer 1 Report

This is the comments on the Manuscript submitted to Materials

Manuscript ID: Materials
Type of manuscript: Article
Title:
Study of high-silicon steel as interior rotor for high-speed mo-2 tor considering the influence of multi-physical field coupling 3 and slotting process

Authors: Deji Ma, Baozhi Tian, Xuejie Zheng, Yulin Li, Shibo Xu and Ruilin,(Department of Electric Engineering Shenyang University of Technology, SUT Shenyang 110870, China 6, Suzhou Inn-Mag New Energy Ltd, Inn-mag Suzhou 215000, China).
Submitted to section:
Electronic Materials

Rate the Manuscript:

1. Significance to field and specialization of “Materials” journal: good.

The paper contains the discussion about in trend of high-speed motor increase power density. The surface-mounted structure is commonly used for high-speed motor rotors, but the performance deterioration caused by the rotor sheath is quite obvious. Compared with the surface-mounted structure, the interior rotor structure has features such as high power density and high efficiency. High-silicon steel (10JNEX900) has low loss and high yield strength and is suitable as rotor core for high-speed motor. Therefore, in this paper, the magnetization and loss characteristics of high silicon steel are tested and analyzed under normal and multi-physical fields. Meanwhile, a interior rotor structure of high-speed motor using high silicon steel as rotor core is proposed due to the test results. The rotor was also subjected to multi-physics field simulation analysis of electromagnetic, mechanical and thermal properties. Considering the brittleness of high-silicon steel, an experimental comparative analysis was conducted in terms of the slotting process of the core.

2. Scientific content:   good.

3. Originality: good.

4. Clarity and presentation:  acceptable.

5. Appropriateness for Journal: appropriate subject mater for the “Materials”. Need for rapid publication: no

7. Recommendations: to sent after minor revision to Materials.

Remarks

  • The abstract is not concise. Please, rewrite it (I try do it in chapter 1).
  • Much more should evidence is need before achieve the conclusions.
  • Authors should make sure that they written every sentence to convey their meaning dearly to the not conclusion (I try do it in chapter 1).
  • Article should be written in an organized way.
  • The manuscript should be checked by native speaker for correct grammar and spelling.
  • May be consider as additional reference: Selection of Materials for High-Speed Motor Rotors. Materials Science 38, 293–303 (2002). https://doi.org/10.1023/A:1020910708441

Reviewer 2 Report

Strengths

This paper presents a interior rotor structure of a high-speed permanent magnet synchronous motor with high silicon steel (10JNEX900) as the rotor material. Solves the problem of centrifugal force and loss of  the interior rotor when the motor is running at high speed. To further determine the idea that high silicon steel is suitable for high-speed motor rotors, the magnetic properties of 10JNEX900 were measured in this paper. The results show that the high-silicon steel 10JNEX900 has better magnetic properties than the conventional high-strength steel 20SW1200H in both the normal state and the multi-physical field coupled state.

Meanwhile, considering the actual operating conditions of the high-speed motor rotor, the proposed rotor is analyzed in terms of electromagnetic performance, mechanical performance and thermal performance, and the analysis results of the 10JNEX900 rotor and the 20SW1200 rotor are compared and analyzed in this paper. The results show that motors with high-silicon steel rotors have higher efficiency, safer performance and lower temperature rise than rotors with conventional high-strength steel.   

Weakness    

1. Citation are not filled in (Line 24).

2. References to Figures 2, 7, 9, 10, 11, 13, 14, 19  after their appearance in the text.

3….  and the highest temperature point is 3.332°C higher. (Line 278). Figure 15. What is the physical meaning of  the temperature values presented with such high accuracy?

4. Figure 13. Axis designations in Chinese.

5. Only one publication  [22] of  the authors in the references.

Reviewer 3 Report

Some questions/comments/suggestions for paper improvement are as follows:

(1) There are considered the eddy current losses in the permanent pagnets in the evaluation of the motor efficiency?

(2) The maximum value of the temperature considered in the paper, 100 Celsius degree it is not relatively low ?

(3) For better visualization of the differences I propose a unique legend in the figure which contains two color maps/images.

(4) Explanation related to the difference between the two color maps in Figure 9 in the four faces of the permanent magnets !!!
